# FLS2 is a CDK-like kinase that directly binds IFT70 and is required for proper ciliary disassembly in *Chlamydomonas*

**Qin Zhao**[1], **Shufen Li**[1], **Shangjin Shao**[1], **Zhengmao Wang**[1], **Junmin Pan**[1,2] *

1 MOE Key Laboratory of Protein Sciences, Tsinghua-Peking Center for Life Sciences, School of Life Sciences, Tsinghua University, Beijing, China, 2 Laboratory for Marine Biology and Biotechnology, Qingdao National Laboratory for Marine Science and Technology, Qingdao, Shandong Province, China

* panjunmin@tsinghua.edu.cn

## Abstract

Intraflagellar transport (IFT) is required for ciliary assembly and maintenance. While disruption of IFT may trigger ciliary disassembly, we show here that IFT mediated transport of a CDK-like kinase ensures proper ciliary disassembly. Mutations in flagellar shortening 2 (*FLS2*), encoding a CDK-like kinase, lead to retardation of cilia resorption and delay of cell cycle progression. Stimulation for ciliary disassembly induces gradual dephosphorylation of FLS2 accompanied with gradual inactivation. Loss of FLS2 or its kinase activity induces early onset of kinesin13 phosphorylation in cilia. FLS2 is predominantly localized in the cell body, however, it is transported to cilia upon induction of ciliary disassembly. FLS2 directly interacts with IFT70 and loss of this interaction inhibits its ciliary transport, leading to dysregulation of kinesin13 phosphorylation and retardation of ciliary disassembly. Thus, this work demonstrates that IFT plays active roles in controlling proper ciliary disassembly by transporting a protein kinase to cilia to regulate a microtubule depolymerizer.

## Author summary

Cilia or eukaryotic flagella are cellular surface protrusions that function in cell motility as well as sensing. They are dynamic structures that undergo assembly and disassembly. Cilia are resorbed during cell cycle progression. Dysregulation of cilia resorption may cause delay of cell cycle progression, which underlies aberrant cell differentiation and even cancer. Ciliary resorption requires depolmerization of axonemal microtubules that is mediated by kinesin13. Using the unicellular green alga, *Chlamydomonas*, we have identified a CDK-like kinase FLS2 that when mutated retards cilia resorption, leading to delay of cell cycle progression. FLS2, a cell body protein, is transported to cilia via intraflagellar transport upon induction of cilia resorption. FLS2 directly interacts with IFT70 and loss of this interaction inhibits transport of FLS2 to cilia and fails to regulate proper phosphorylation of kinesin13 in cilia.

**Data Availability Statement:** All relevant data are within the manuscript and its Supporting Information files.

**Funding:** This work was supported by the National Key R&D Program of China (2017YFA0503500,

2018YFA0902500) and the National Natural Science Foundation of China (31671387 and 31972888) to JP. The funders had no role in study design, data collection and analysis, decision to publish, or preparation of the manuscript.

**Competing interests:** The authors have declared that no competing interests exist.

## Introduction

Cilia are microtubule-based cellular structures that extend from the cell surface. The cellular motility and signaling mediated by cilia plays pivotal roles in physiology and development [1, 2]. The medical importance of cilia is underscored by at least 35 ciliopathies that are caused by mutations in around 200 cilia-related genes [3].

Cilia are dynamic structures that undergo assembly and disassembly. They are assembled after cell division and disassembled prior to and/or during mitosis [4–7]. They are also subjected to disassembly during cell differentiation and in response to cellular stress [8–10]. Ciliary disassembly may occur via deflagellation/deciliation (shedding of whole cilium or flagellum) or resorption (gradual shortening from the ciliary tip) depending on physiological context and/or stimulus [8, 11, 12]. During cell cycle progression in mammalian cells as well as in *Chlamydomonas*, cilia are resorbed [4, 5, 7, 13, 14]. However, deciliation has also been reported as a predominant mode of ciliary disassembly during cell cycle progression in mammalian cells [15]. Defect in ciliary resorption has been shown to inhibit G1-S transition and delays cell cycle progression [16–20], which leads to premature differentiation of neural progenitors [18, 20, 21]. Several studies also suggest that ciliary disassembly is related to tumorigenesis because primary cilia are disassembled in a variety of cancer types [22].

Cilia resorption is triggered by a series of signal cascades that almost all lead to activation of aurora-A [7, 10, 23]. Aurora-A further activates microtubule deacetylase HDAC6 and inhibition of which impairs ciliary disassembly [7]. Microtubule depolymerizing kinesins also function in ciliary resorption to mediate disassembly of axonemal microtubules. Depletion of microtubule depolymerases kinesin13s in *Chlamydomonas* (CrKinesin13) or mammalian cells (KIF2A and KIF24) inhibits ciliary disassembly [20, 24–26].

Intraflagellar transport (IFT) ferries ciliary proteins to build and maintain cilia [27, 28]. Conditional inactivation of IFT motor kinesin-2 induces ciliary disassembly in both mammalian cells and in *Chlamydomonas* [23, 29–31], suggesting that cells may employ the mechanism of inactivation of IFT machinery to trigger ciliary disassembly. However, it has been shown that IFT proteins are actually increased in resorbing cilia upon cilia resorption that occurs during zygote development or in response to extracellular stimuli in *Chlamydomonas* [32, 33], implying that IFT may be involved in ciliary resorption triggered by internal or external stimuli. Thus, further evidence is needed to pinpoint the role of IFT in ciliary resorption.

In this report, we have identified a CDK-like kinase, flagellar shortening 2 (FLS2) that functions in ciliary resorption in *Chlamydomonas*. FLS2 promotes ciliary shortening by a mechanism in which it is transported to cilia by directly binding IFT70 to control proper phosphorylation of kinesin13 in cilia.

## Results

### Ciliary resorption is impaired in an *fls2* mutant that was defective in a gene encoding a CDK-like kinase

Addition of sodium pyrophosphate (NaPPi) to the *Chlamydomonas* cell cultures induces gradual cilia shortening or cilia resorption but not deciliation [23, 34], which provides an excellent system to screen for mutants defective in cilia resorption. In this study, we generated an *Aph-VIII* DNA insertional mutant, termed flagellar shortening 2 (*fls2)*, which underwent a slow kinetics of ciliary shortening in contrast to wild type cells (Fig 1A and 1B, S1 Table). The steady state ciliary length in the mutant and wild type cells was similar (Fig 1A), indicating that the mutant is only defective in ciliary disassembly but not assembly or ciliary length control. The foreign DNA insert was localized in the last exon of an unknown gene (Cre03.g169500), which

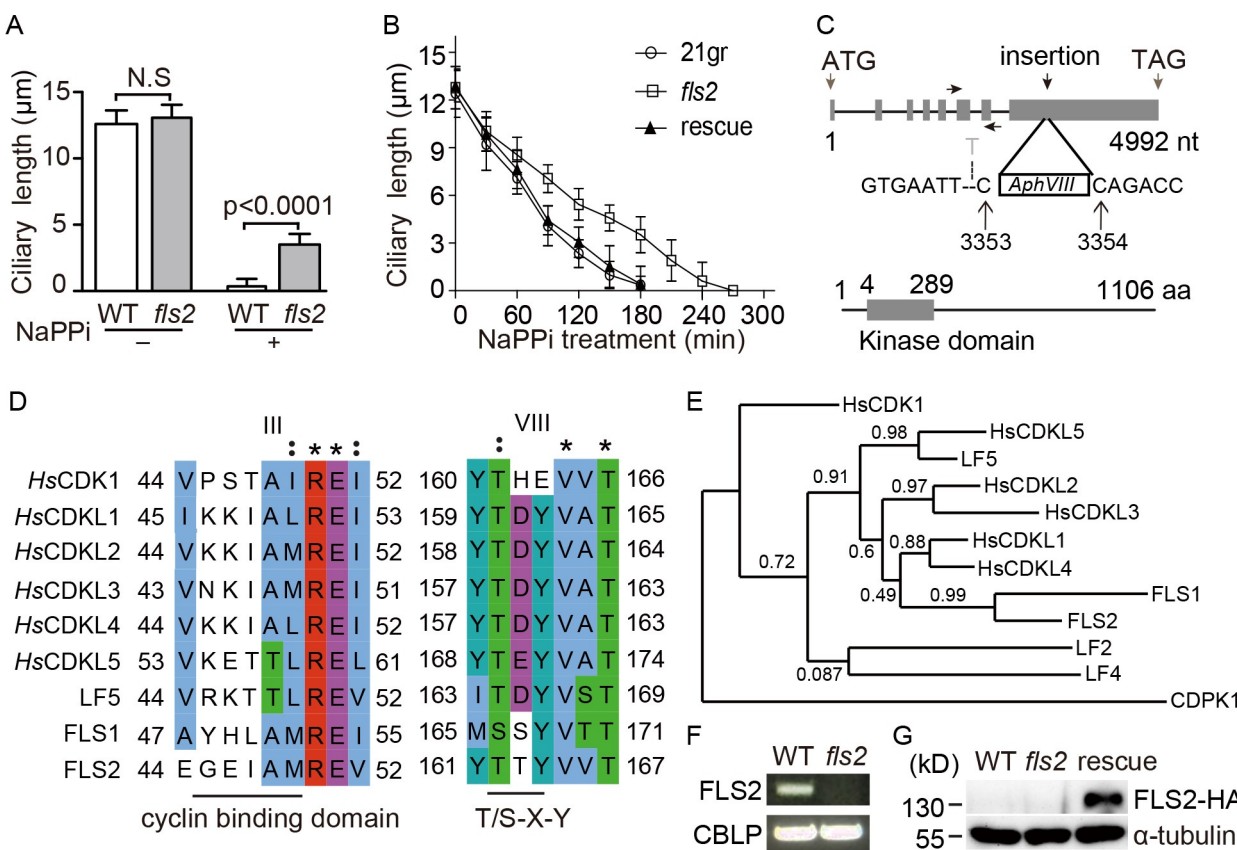

**Fig 1. Identification of a CDK-like kinase, FLS2 that functions in ciliary disassembly.** (A) *fls2* mutant is defective in cilia resorption but not steady state ciliary length. Wild type (WT) or *fls2* cells were treated with or without 20 mM NaPPi for three hours followed by ciliary length measurement. Ciliary length data shown here and below are presented as mean±SD with n = 50 cilia. N.S., not significant (*p*>0.05). (B) *fls2* mutant exhibits slower kinetics of ciliary disassembly. WT, *fls2* and the rescued cells were induced for ciliary disassembly by addition of 20 mM NaPPi. Ciliary length was measured at the indicated times. (C) Diagrams of the gene structure of *FLS2* with the indicated foreign DNA insertion site and the domain structure of the protein kinase encoded by *FLS2*. The *AphVIII* DNA fragment is inserted in the last exon of *FLS2* between 3353 and 3354 nt and results in deletion of the 3352 nt. The left and right arrows show the positions of the primers used for RT-PCR. (D) Alignment of the protein kinase domain III and VIII of FLS2 with those of human CDK1, CDK-like kinases (CDKLs) and two *Chlamydomonas* CDKLs (LF5 and FLS1) that are implicated in ciliary functions. (E) Phylogenetic analysis places FLS2 in the group of CDKLs. A neighbor-joining phylogenetic tree was constructed by using an algorithm (www.phylogeny.fr) following the instruction. FLS2 was analyzed with the human CDKLs and two *Chlamdyomonas* CDKLs: FLS1 and LF5. The outgroup members include LF2 and LF4, two MAPK-related kinases in *Chlamdyomona*s; CDPK1, a *Chlamydomonas* calcium dependent kinase and HsCDK1, a cyclin-dependent kinase. The numbers above the line indicate the bootstrap values. The sequences of the kinase domains were used for the analysis. (F) *FLS2* is not expressed in *fls2* cells shown by RT-PCR. Gene expression of *CBLP* was used as a control. (G) An immunoblot showing expression of *FLS2-HA* in *fls2* cells. WT and *fls2* cells were used as controls.

encodes a protein kinase of 1106 aa (Fig 1C). FLS2 is similar to CDK-like protein kinases with a putative cyclin binding domain and a T/S-X-Y motif at the kinase activation loop [35, 36] (Fig 1D). As expected, phylogenetic analysis has placed FLS2 into the group of CDK-like kinases (Fig 1E).

To determine whether *FLS2* expression was disrupted in the mutant, we attempted to make antibodies but it was unsuccessful. However, RT-PCR showed that *FLS2* transcript was not detected (Fig 1F), indicating that foreign DNA insertion likely causes decay of *FLS2* mRNAs. To confirm that disruption of *FLS2* is indeed responsible for the observed ciliary phenotype, HA-tagged *FLS2* was expressed in *fls2* (Fig 1G). As expected, ciliary shortening defect of *fls2* was rescued (Fig 1B). Thus, we have identified a CDK-like kinase, FLS2, which functions in ciliary disassembly.

## FLS2 regulates ciliary disassembly under physiological conditions and cell cycle progression

To examine whether *fls2* mutation also affects ciliary disassembly under physiological conditions, we first analyzed ciliary shortening during zygotic development [14, 32]. To generate zygotes in *fls2* background, we isolated an mt- strain of *fls2* by crossing the original mt+ *fls2* strain with a wild type mt- strain. As shown in Fig 2A(S1 Table), ciliary disassembly in *fls2* zygotes was retarded as compared to the wild type control. *Chlamydomonas* cells also disassemble their cilia via gradual ciliary shortening during cell cycle progression [13, 14]. To examine ciliary disassembly in *fls2* during cell cycle progression, cells were synchronized by a light: dark (14h:10h) cycle. Ciliary length was measured during cell cycle progression. As shown in Fig 2B (S1 Table), ciliary disassembly in *fls2* was retarded as compared to the control. Defects in ciliary disassembly during G1 to S transition has been shown to delay cell cycle progression in mammalian cells as well as in *Chlamydomonas* [13, 16–18]. As expected, *fls2* mutant showed a delay of cell cycle progression (Fig 2C, S1 Table). Thus, FLS2 is involved in ciliary disassembly under physiological and non-physiological conditions and defects in *FLS2* has physiological consequences.

## The kinase activity of FLS2 is required for proper ciliary disassembly and gradually down-regulated by dephosphorylation

Proteins may exhibit gel mobility shift in SDS-PAGE due to changes in the state of protein phosphorylation. To detect possible changes in FLS2 phosphorylation, we analyzed rescued cells expressing FLS2-HA during ciliary shortening induced by NaPPi by immunoblotting. FLS2-HA did not show gel motility shift during the entire course of ciliary disassembly (Fig 3A). The cell lysates were then analyzed in Phos-tag SDS-PAGE followed by immunoblotting, which has a better separation of phosphoproteins [37, 38]. Before NaPPi treatment (time 0), FLS2-HA exhibited apparently two migrating forms (Fig 3B). The slower migrating form of FLS2-HA gradually disappeared during ciliary disassembly, indicating that FLS2 is a phosphoprotein in steady state cells and gradually dephosphorylated. Phosphatase treatment confirmed that the gel mobility shifts of FLS2-HA were indeed caused by phosphorylation (Fig 3B).

To examine the relationship between FLS2 phosphorylation and its kinase activity, FLS2-HA was immunoprecipitated from control cells and cells that underwent ciliary disassembly for 180 min followed by *in vitro* kinase assay. The kinase activity of FLS2-HA was greatly decreased at 180 min, indicating that the phosphorylation state of FLS2 correlates with its kinase activity (Fig 3C). To test whether the kinase activity of FLS2 is required for ciliary disassembly, a kinase-dead version of *FLS2* (*K33R*) tagged with HA was expressed in *fls2* cells. *In vitro* kinase assay showed that *K33R* mutant barely had any kinase activity (Fig 3C). Examination of the ciliary disassembly kinetics of *K33R* mutant showed that ciliary disassembly defect in *fls2* was not rescued (Fig 3D, S1 Table). Taken together, these data suggest that FLS2 is an active kinase and gradually down-regulated due to dephosphorylation during ciliary disassembly and its kinase activity is required for proper ciliary disassembly.

## FLS2 functions in suppressing phosphorylation of CrKinesin13 during early stages of ciliary resorption

Next, we were interested in understanding the working mechanism of FLS2 during ciliary disassembly. We examined several factors that have previously been implicated in ciliary disassembly. *Chlamydomonas* aurora-like kinase CALK, which is a homologue of aurora-A, is phosphorylated during and required for ciliary disassembly [7, 23, 39]. To examine whether

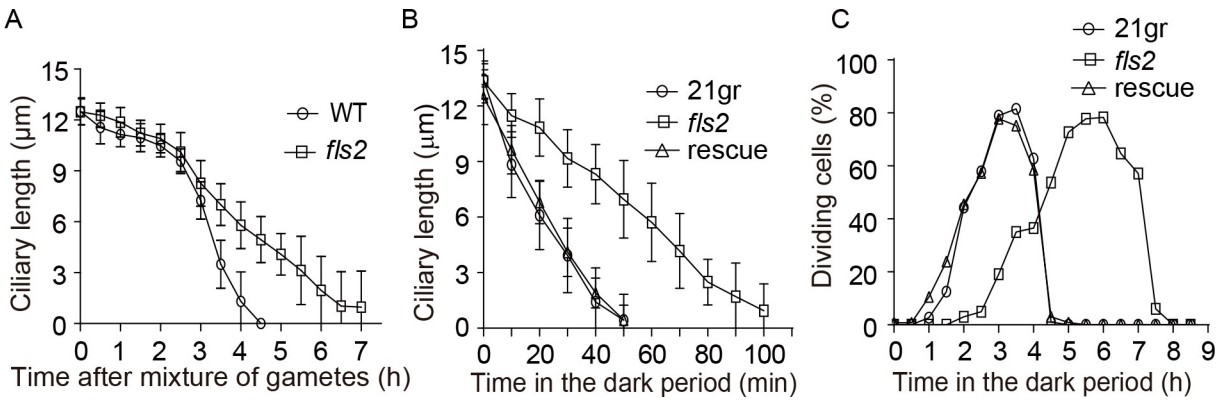

**Fig 2. *FLS2* regulates ciliary disassembly under physiological conditions and cell cycle progression.** (A) *fls2* is defective in ciliary disassembly during zygote development. Gametes of opposite mating types of *fls2* were mixed to allow mating and zygote development followed by ciliary length measurement. The mating of wild type gametes was used as a control. Error bars indicate SD. (B) *fls2* is defective in ciliary disassembly during cell cycle progression. WT, *fls2* and rescued cells were synchronized by a light/dark (14h/10h) cycle. Ciliary length was scored beginning in the dark period when cells prepared to enter mitosis. (C) Cell cycle progression in *fls2* mutant is retarded. Dividing cells were scored microscopically in the dark period of the light/dark cycle. Compared to WT and rescued cells, the peak of cell division in *fls2* mutant was delayed around 3 hrs.

FLS2 regulates CALK phosphorylation, wild type (WT) and *fls2* cells were induced to shorten their cilia by NaPPi treatment followed by immunoblotting. CALK phosphorylation in *fls2* was not affected as compared to the control (Fig 4A). Increased trafficking of IFT proteins into

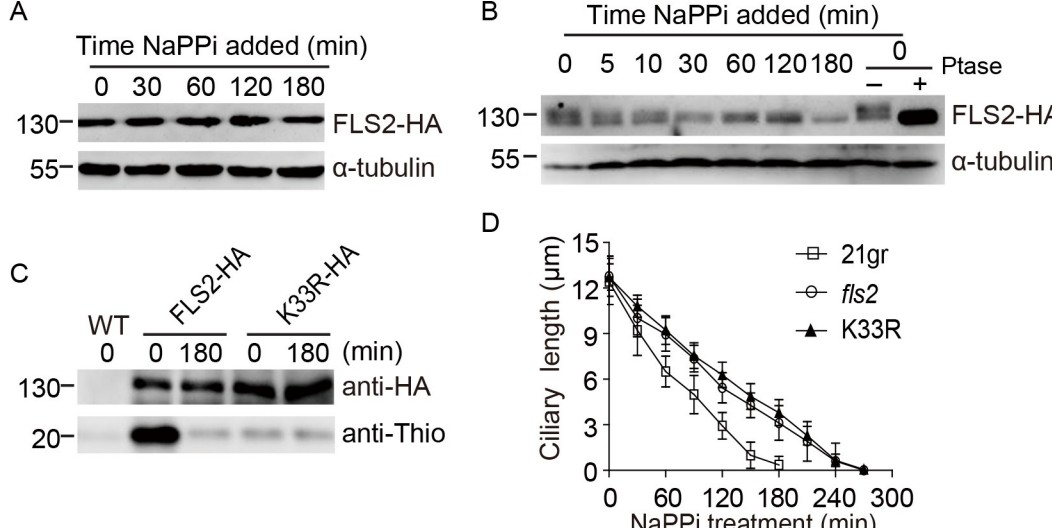

**Fig 3. FLS2 kinase activity is required for ciliary disassembly during which it undergoes dephosphorylation and inactivation.** (A) *FLS2* expression is not altered during ciliary disassembly. *fls2* rescued cells expressing *FLS2-HA* was induced for ciliary disassembly by addition of NaPPi followed by immunoblotting with the indicated antibodies. (B) FLS2 undergoes gradual dephosphorylation during ciliary disassembly. A phos-tag immunoblotting was performed using antibodies as indicated. Steady state cells were treated with or without phosphatase (Ptase) to demonstrate that the gel mobility shift was caused by changes in protein phosphorylation. (C) Dephosphorylation of FLS2 results in loss of its kinase activity. *fls2* cells expressing wild type *FLS2-HA* or kinase dead mutant *K33R-HA* were treated with 20 mM NaPPi for the indicated times. Wild type (WT) cells were used as a negative control. Cell lysates were incubated with anti-HA antibodies for immunoprecipitation followed by *in vitro* kinase assay. ATPγS was used as ATP donor and myelin basic protein as substrate. Anti-thiophosphate ester antibody was used to detect substrate phosphorylation. (D) The kinase activity of FLS2 is required for ciliary disassembly. Cells from WT, *fls2* and kinase-dead mutant *K33R* were induced for ciliary disassembly by NaPPi treatment. Ciliary length was measured at the indicated times. Bars indicate SD.

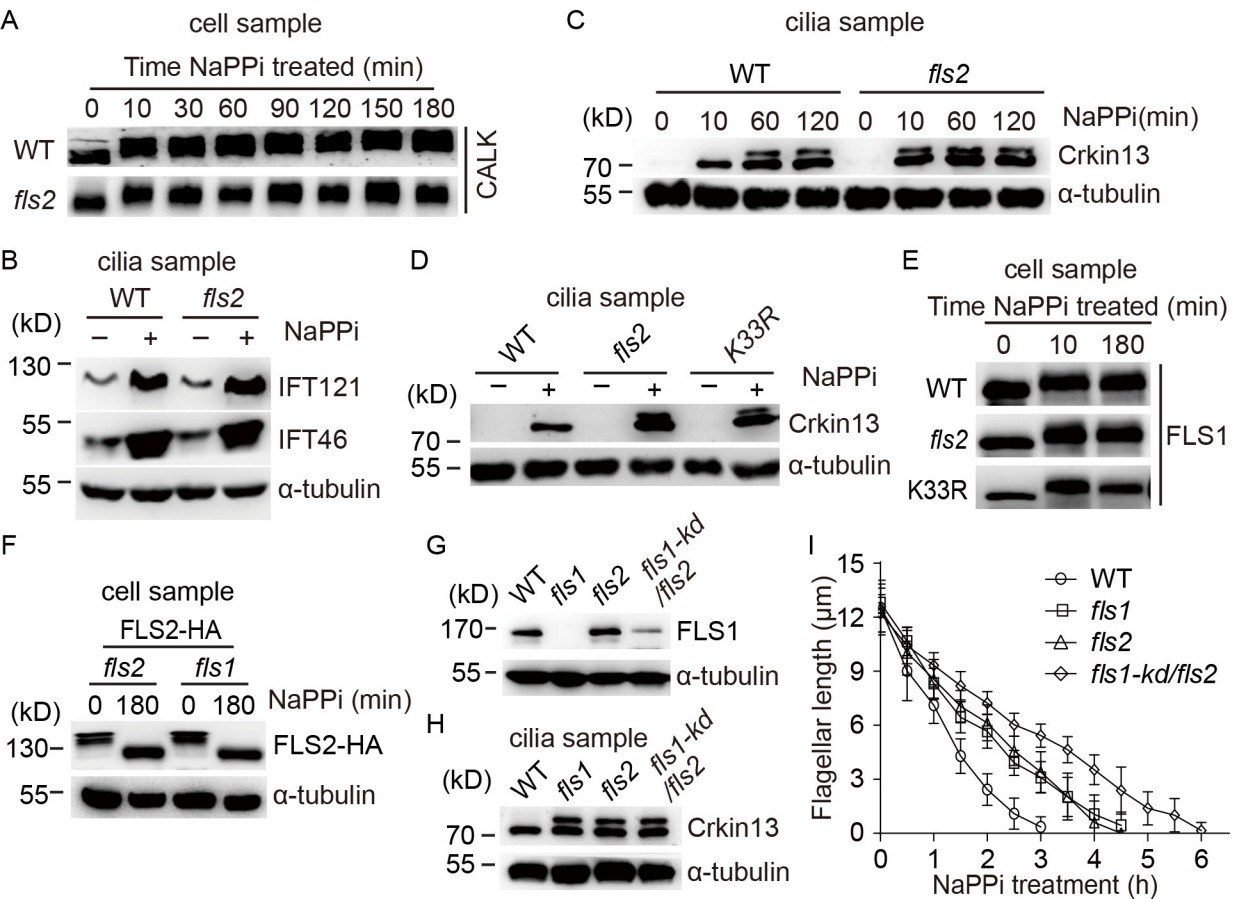

**Fig 4. FLS2 does not regulate CALK and IFT but suppresses CrKinesin13 phosphorylation independently of FLS1.** (A) CALK phosphorylation is not affected in *fls2*. WT and *fls2* cells were treated with NaPPi for ciliary disassembly. Cell lysates at the indicated time were analyzed by immunoblotting with CALK antibody. CALK underwent phosphorylation with slower gel migration, which were not affected in *fls2*. (B) FLS2 does not affect increasing transport of IFT proteins in cilia. WT or *fls2* Cells were treated with or without 20 mM NaPPi for 10 min followed by cilia isolation and immunoblotting. IFT121, a subunit of IFT-A and IFT46, a subunit of IFT-B were analyzed. α-tubulin was used as a loading control. (C) Loss of FLS2 does not affect ciliary transport of CrKinesin13 but induces earlier onset of its phosphorylation. WT or *fls2* cells were treated with NaPPi for the indicated times followed by cilia isolation and immunoblotting. The slower migration form of CrKinesin13 is due to phosphorylation as demonstrated previously. (D) The kinase activity of FLS2 is required for suppression of earlier onset of CrKinesin13 phosphorylation. Isolated cilia from cell samples that were treated with or without NaPPi for 10 min were subjected to immunoblot analysis with the indicated antibodies. (E) FLS1 phosphorylation is not altered in *fls2*. Cell samples as indicated were analyzed by immunoblotting with FLS1 antibody. (F) FLS2 phosphorylation is not altered in *fls1*. *fls1* or *fls2* cells expressing FLS2-HA were treated for the indicated times followed by Phos-tag immunoblotting. FLS2 phosphorylation as evidenced by patterns of gel migration was similar between *fls1* and *fls2* cells. (G) Generation of an *fls1-kd/fls2* strain by RNAi depletion of FLS1 in *fls2* cells. Cell samples as indicated were subjected to immunoblotting with anti-FLS1 antibody. (H) CrKinesin13 phosphorylation in *fls1-kd/fls2* cells. Cilia were isolated from cell samples that were treated with NaPPi for 10 min and were then subjected to immunoblotting. (I) *fls1-kd/fls2* cells show more severe defect in ciliary disassembly. Cells as indicated were induced for ciliary disassembly followed by ciliary length measurement at the indicated times. Bars indicate SD.

cilia occurs during ciliary disassembly and the function of which is not clear [13, 32, 40]. Representative IFT proteins IFT121 (IFT-A) and IFT46 (IFT-B) were similarly increased in WT and *fls2* cilia isolated from cells undergoing ciliary disassembly (Fig 4B). Thus, FLS2 is not involved in CALK phosphorylation and increased ciliary trafficking of IFT proteins.

A single kinesin13 is present in *Chlamdyomonas* and has been shown to be required for ciliary resorption [25]. It is transported from the cell body to cilia upon stimulation for ciliary disassembly and becomes partially phosphorylated at 60 min after NaPPi treatment [13, 25] (Fig 4C). We therefore examined whether FLS2 affects the behaviors of CrKinesin13. Immunoblot analysis showed that loss of FLS2 did not affect ciliary transport of CrKinesin13 but lead to

early onset of CrKinesin13 phosphorylation already 10 min after adding NaPPi (Fig 4C). The kinase activity of FLS2 was required for this inhibition as demonstrated by using a kinase-dead mutant *K33R* (Fig 4D). These data suggest that at least one of the mechanisms of FLS2 in regulation of ciliary disassembly is to suppress CrKinesin13 phosphorylation during the early stage of ciliary resorption.

Previously, we have shown that loss of FLS1 also induces earlier onset of CrKinesin13 phosphorylation and slows ciliary disassembly [13]. Thus, it raises a question whether FLS1 and FLS2 affect each other. FLS1 is phosphorylated upon induction of ciliary disassembly. Immunoblot analysis showed that FLS1 phosphorylation in *fls2* or *K33R* was not altered upon induction of ciliary disassembly (Fig 4E), indicating that FLS2 does not regulate FLS1 phosphorylation. As shown above in Fig 3B, FLS2 undergoes dephosphorylation during ciliary disassembly. We examined whether FLS1 affects FLS2 dephosphorylation. To do this, *fls1* cells expressing FLS2-HA were induced for ciliary disassembly followed by Phos-tag immunoblotting. FLS2-HA in *fls1* cells showed similar dephosphorylation relative to the control (Fig 4F), indicating that FLS1 is not involved in FLS2 dephosphorylation. Thus, the above data suggest that FLS1 and FLS2 do not affect protein phosphorylation of each other.

It was intriguing to learn the ciliary shortening phenotype of an *fls1/fls2* double mutant. However, it was unsuccessful to obtain such a mutant by crossing *fls1* and *fls2*. We decided to deplete FLS1 in *fls2* cells by RNAi as our attempt to knock out *FLS1* by CRISPR/Cas9 failed. RNAi resulted in approximately 80% deletion of FLS1 expression in *fls1-kd/fls2* cells (Fig 4G). CrKinesin13 was phosphorylated in disassembling cilia of *fls1-kd/fls2* but the degree of its phosphorylation was similar to that in *fls1* or *fls2* single mutant (Fig 4H), indicating that FLS1 and FLS2 act in the same pathway to regulate phosphorylation of CrKinesin13. Examination of ciliary disassembly found that *fls1-kd/fls2* cells showed more severe defect in ciliary disassembly than *fls1* or *fls2* alone (Fig 4I, S1 Table), suggesting critical roles of FLS1 and FLS2 in ciliary disassembly. Please note, *fls1* showed a constant slow rate of ciliary disassembly that is contrast to the previous report where *fls1* exhibited a slow rate of disassembly followed by a relatively faster rate [13]. The reason is not clear.

## FLS2 is localized in the cell body and undergoes increased ciliary trafficking during ciliary disassembly

To learn more about how FLS2 functions in ciliary disassembly, we determined the cellular distributions of FLS2-HA. Immunostaining showed that FLS2-HA predominantly localized in the cell body (Fig 5A), which was confirmed by immunoblotting of isolated cell bodies and cilia (Fig 5B). Interestingly, upon induction of ciliary disassembly, FLS2-HA was transported to cilia (Fig 5A and 5B). FLS2-HA in cilia was associated with the axoneme (Fig 5C). To determine whether transport of FLS2 into cilia upon induction of ciliary disassembly also occurs under physiological conditions, we examined ciliary disassembly during zygotic development. Immunostaining analysis showed that FLS2-HA also increased in cilia of zygotes that underwent ciliary disassembly (Fig 5D).

The increase of FLS2 in cilia was rapid. As early as 10 min after induction of ciliary disassembly, the increase of FLS2-HA was detected (Fig 5C and 5E). The increased amounts of FLS2-HA in cilia between 10 and 120 min after induction of ciliary disassembly was similar, one possible explanation is that the increase in ciliary FLS2 might be caused by increased ciliary trafficking of IFT rather than by FLS2 accumulation (Fig 5E). The continuous ciliary trafficking of FLS2 during ciliary disassembly is also supported by the data that the ciliary form of FLS2-HA showed different phosphorylation states during ciliary disassembly, which is similar to the whole cell form of FLS2-HA (Fig 5E and Fig 3B). This data also suggests that the

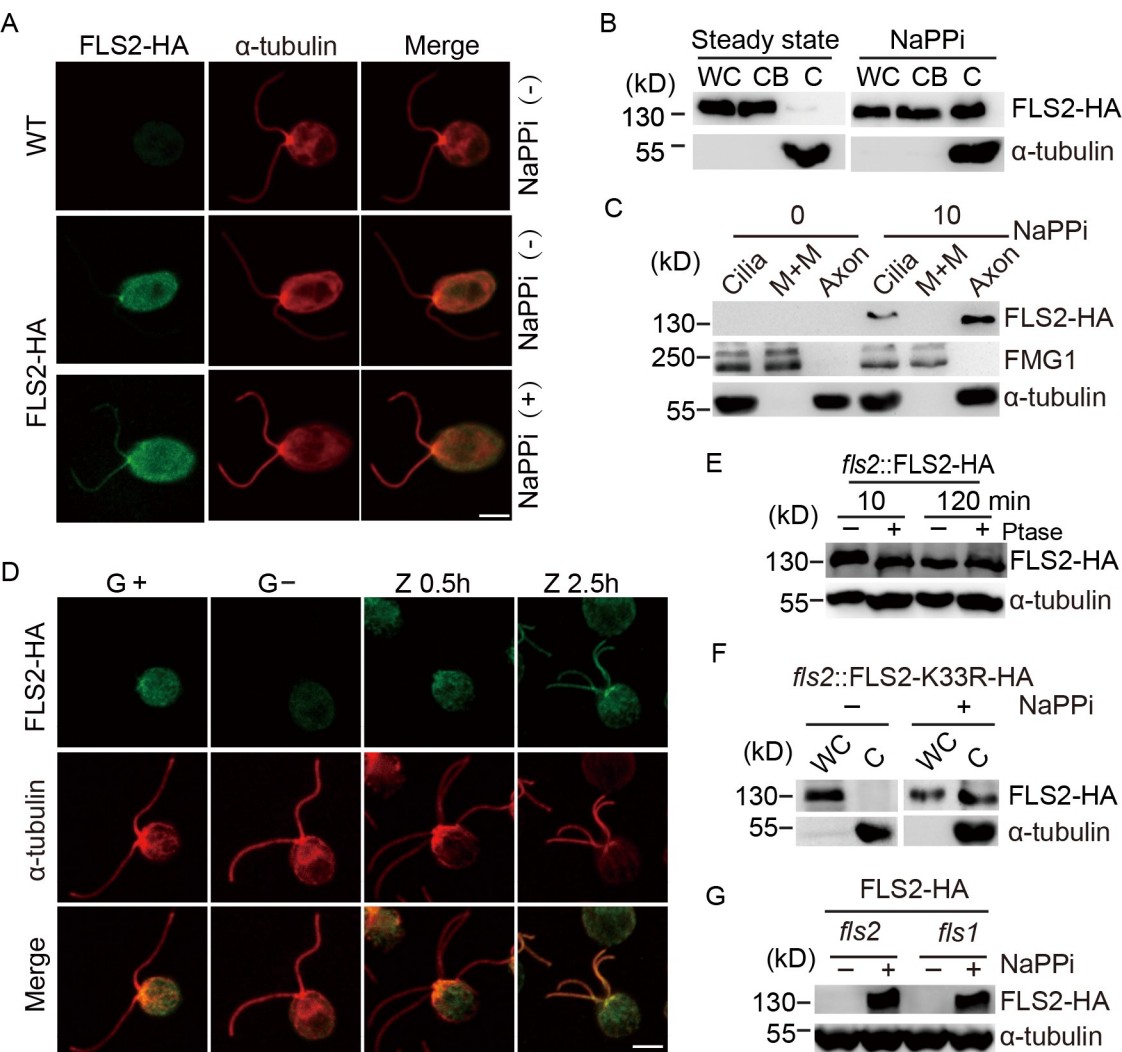

**Fig 5. Increased ciliary trafficking of FLS2 upon induction of ciliary disassembly.** (A) FLS2 is a cell body protein and transported to cilia during ciliary disassembly as examined by immunostaining. *fls2* cells expressing *FLS2-HA* were treated with or without NaPPi for 10 min followed by immunostaining with HA and α-tubulin antibodies. WT cells were used as control. Bar, 5 μm. (B) Analysis of ciliary transport of FLS2 by immunoblotting. *fls2* cells expressing *FLS2-HA* were separated into cell bodies (CB) and cilia (C) fractions after treatment with or without NaPPi for 10 min followed by immunoblotting. WC, whole cells. (C) Ciliary FLS2 is associated with the axonemes. Cilia isolated from cells treated with or without NaPPi for 10 min were fractionated into membrane matrix (M+M) and axonemal fractions followed by immunoblotting with the indicated antibodies. FMG1, a ciliary membrane protein was used as a control for M+M fractions. (D) FLS2 is transported to cilia during ciliary disassembly in zygote development. Immunostaining analysis of FLS2 in mt+ and mt- gametes (G+ and G-), 0.5 hr (Z0.5h) and 2.5 hr (Z2.5h) zygotes. Bar, 5 μm. (E) FLS2 in cilia undergoes dephosphorylation during ciliary disassembly but its levels are unchanged. Cilia were isolated from cells treated with NaPPi for 10 or 120 min. The isolated cilia were treated with or without phosphatase (Ptase) followed by phos-tag immunoblotting. Please note that FLS2 in the 10 min sample without phosphatase treatment exhibited slower gel motility shift relative to other samples. (F) The kinase activity of FLS2 is not required for its ciliary transport. Whole cells (WC) or isolated cilia from kinase-dead *K33R* mutant cells treated with or without NaPPi for 10 min were subjected to immunoblot analysis. (G) FLS1 does not affect ciliary transport of FLS2. Cilia isolated from *fls1* cells expressing FLS2-HA treated with or without NaPPi were analyzed by phos-tag immunoblotting. *fls2* cells expressing *FLS2-HA* were used as control. Ciliary transport as well as gel mobility of FLS2 expressed in *fls1* were similar to the control.

phosphorylation state of FLS2 is not involved in ciliary trafficking of FLS2. We further showed that K33R mutant of FLS2 was able to be transported to cilia (Fig 5F). Thus, FLS2 undergoes increased ciliary trafficking upon induction of ciliary disassembly and its phosphorylation

state and kinase activity are not required for this process. We further showed that ciliary trafficking of FLS2-HA in *fls1* was not affected (Fig 5G), indicating that FLS1 does not regulate ciliary trafficking of FLS2.

## FLS2 is a cargo of IFT70

Though ciliary proteins may diffuse to cilia, IFT appears to be the major mechanism for transporting ciliary proteins into cilia [41–43]. To determine whether IFT was required for FLS2 transport, we took advantage of the temperature conditional kinesin-2 mutant *fla10-1*, in which IFT gradually diminishes at the non-permissive temperature [44, 45]. *fla10-1* cells expressing FLS2-HA were incubated at 22 or 32 $^{o}$C for 1 hour followed by addition of NaPPi for 10 min to stimulate ciliary disassembly. The cilia were then isolated for immunoblot analysis. As shown in Fig 6A, IFT139, FLS2-HA as well as CrKinesin13 were increased in cilia at 22 $^{o}$C after stimulation by NaPPi. At 32˚C, however, IFT protein IFT139 was depleted, and FLS2-HA and CrKinesin13 failed to increase in cilia after NaPPi stimulation. Thus, these results indicate that the transport of FLS2 into cilia depends on IFT.

To identify which IFT proteins may act as cargo adaptors for transport of FLS2, we performed yeast two-hybrid screening by using FLS2 as a bait and each subunit of the IFT complex (IFT-B and IFT-A) as a prey. FLS2 interacted only with IFT70 (Fig 6B). To determine the minimal segment of FLS2 required for this interaction, a series of deletion mutants were probed for interaction with IFT70 by yeast two-hybrid assay (Fig 6C). The C-terminal region apart from the kinase domain interacted with IFT70 while the N-terminal region containing the kinase domain did not. Interestingly, various C-terminal segments all showed interaction with IFT70 but with reduced capacity. This data may suggest that the C-terminal non-kinase region as a whole is required for tight interaction with IFT70. To further verify this interaction, a GST pull-down assay was performed. IFT70 was pulled down by the C-terminus of FLS2 (FLS2-CT) tagged with GST (Fig 6D). Finally, we showed that IFT70 was co-immunoprecipitated with FLS2-HA (Fig 6E).

IFT70 is a protein with 15 tetratricopeptide repeats (TPRs) [46]. To determine the regions of IFT70 that interact with FLS2, various deletion mutants of IFT70 as indicated were tested for interaction with FLS2 by yeast two-hybrid assay (Fig 6F). It was found that the N-terminal 290 aa region of IFT70 interacted with FLS2. This region has 5 canonical TPRs (no.1-3, 5–6) and two non-canonical TPRs (no.4 and 7) [46]. Deletion of either TPR1, TPR2 or TPR3 abolished the interaction of IFT70 with FLS2 (Fig 6G), indicating that TPR1-3 form a structural module to mediate this interaction.

## Loss of ciliary transport of FLS2 impairs CrKinesin13 phosphorylation and ciliary disassembly

To test whether ciliary transport of FLS2 is required for its function in ciliary disassembly, one strategy would be blocking FLS2 transport by making a mutant of *IFT70* with deletions of TPR1-3. Knockout of *IFT70* in mammalian cells abolishes ciliogenesis, indicating that IFT70 is essential for ciliary assembly [47]. IFT70 tightly interacts with IFT52-IFT88 dimer, which is essential for ciliogenesis [46, 47]. Deletion of TPR1 or TPR1-2 of IFT70 abrogates their interactions with the dimer and could not rescue ciliogenesis in *IFT70* knockout cells [47]. To determine whether TPR1, TPR2 or TPR3 of IFT70 in *Chlamydomonas* functions in interaction with the IFT52-IFT88 dimer, a pull-down assay was performed (Fig 7A). Full-length of IFT70 and TPR4 deletion mutant could interact with the dimer. However, deletion of either TPR1, TPR2 or TPR3 abolished this interaction. As discussed above, we figured that deletion of these TPRs would block ciliary assembly in live cells and it was not feasible to test ciliary transport of FLS2 by using IFT70 deletion mutants.

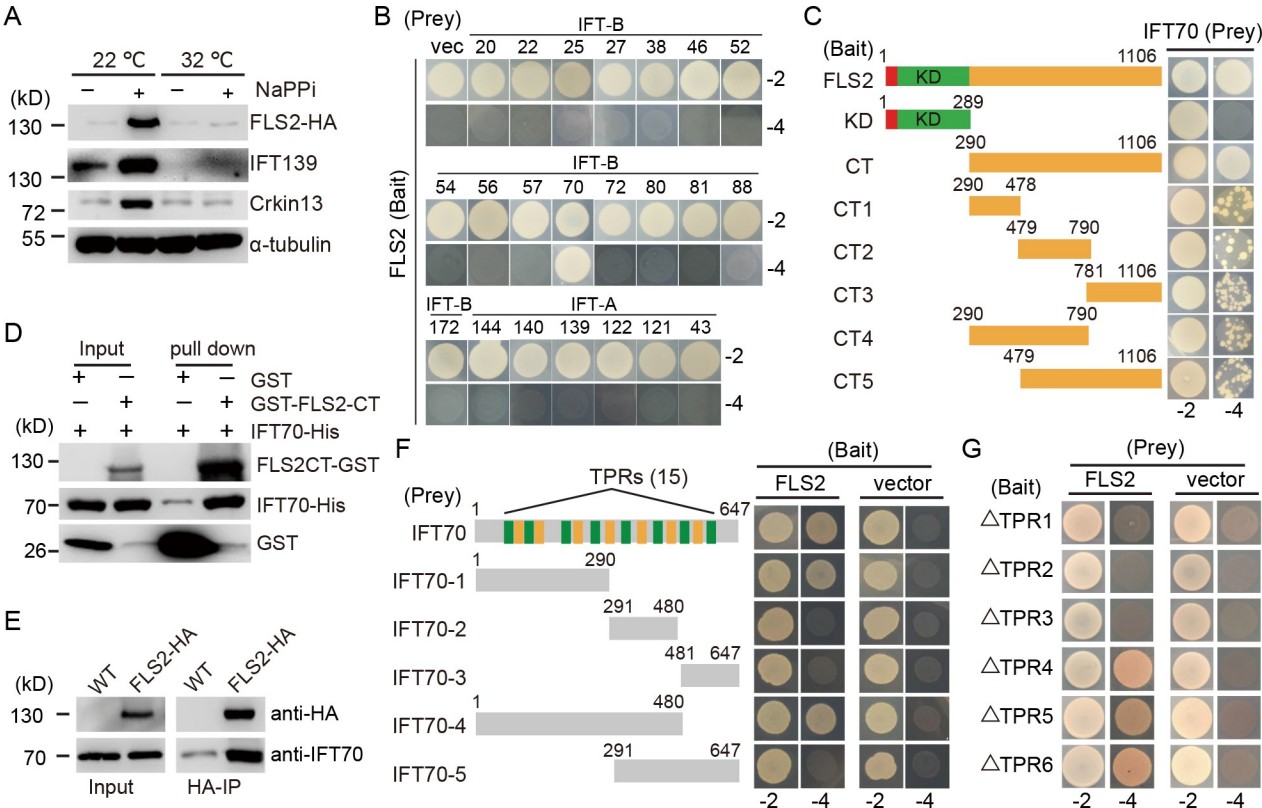

**Fig 6. FLS2 is a cargo of IFT70 for ciliary transport.** (A) Ciliary transport of FLS2 depends on IFT. Temperature sensitive mutant *fla10-1* cells expressing FLS2-HA were incubated at the indicated temperatures for 1h followed by treatment with NaPPi or not for 10 min. The cilia were then isolated for immunoblotting with the indicated antibodies. (B) FLS2 interacts with IFT70 shown in yeast two-hybrid assays. Subunits of IFT-B or IFT-A were transformed respectively into yeast cells with FLS2, followed by growth on selection medium lacking Leu and Trp (-2) or Leu, Trp, His and Ade (-4). (C-D) The C-terminal non-kinase region of FLS2 is required for its interaction with IFT70. Full-length FLS2 and its deletion variants were assayed for their interaction with IFT70 in yeast two-hybrid assays (C). A GST pull-down assay of bacterial expressed FLS2-CT and IFT70 (D). (E) Co-immunoprecipitation of FLS2 and IFT70. Cilia were isolated from control cells (WT) or *fls2* cells expressing FLS2-HA during ciliary disassembly followed by immunoprecipitation with anti-HA antibody and immunoblotting. (F-G) TPR1-3 of IFT70 is essential for its interaction with FLS2. Full-length of IFT70 and its various segments (E) or various TPR deletion mutants (F) were subjected to yeast two-hybrid assays with FLS2.

Because the C-terminal non-kinase domain of FLS2 was required for its interaction with IFT70 (Fig 6C), we decided to delete the C-terminus of FLS2 to see whether it affects ciliary transport of FLS2 and ciliary disassembly. HA-tagged C-terminal truncated mutant (ΔCT) of FLS2 was expressed in *fls2* cells (Fig 7B). Upon induction of ciliary disassembly, the truncated mutant failed to be transported to cilia in contrast to full-length FLS2 (Fig 7C). *In vitro* kinase assay showed that the ΔCT mutant did not affect the kinase activity of FLS2 (Fig 7D), indicating that the C-terminus only functions in ciliary transport. Next, we examined the impact of the C-terminus of FLS2 on CrKinesin13 phosphorylation and ciliary disassembly. Similar to *fls2*, loss of C-terminus of FLS2 failed to suppress CrKinesin13 phosphorylation (Fig 7E) and could not rescue the ciliary disassembly defect of *fls2* (Fig 7F, S1 Table). These data suggest that transport of FLS2 to cilia regulates CrKinesin13 phosphorylation and ciliary disassembly.

## Discussion

In this report, we have identified a CDK-like kinase, namely FLS2 that is involved in cilia resorption by a mechanism in which IFT transports FLS2 between the cell body and cilia to

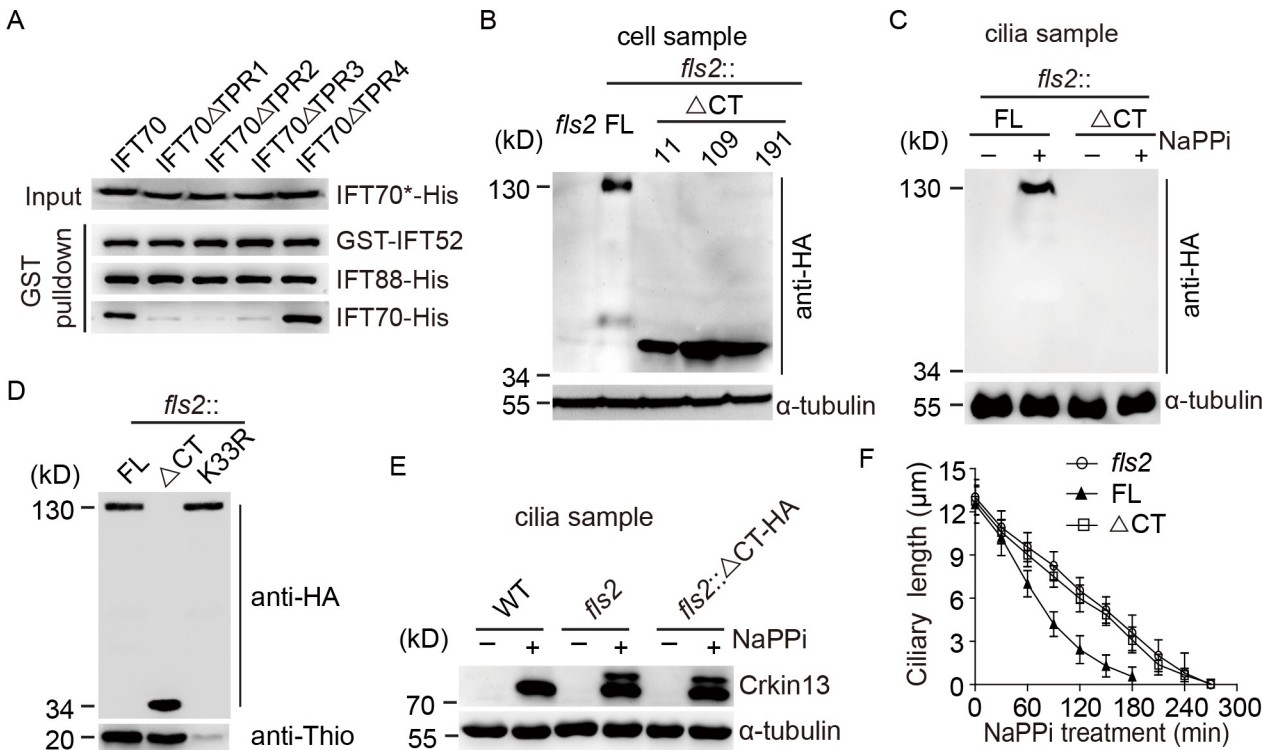

**Fig 7. The C-terminal non-kinase region of FLS2 is required for its ciliary transport and proper ciliary disassembly.** (A) Deletion of TPR1, TPR2 or TPR3 of IFT70 abrogates its interaction with IFT52-IFT88 dimer. Cell lysates from bacterial cells expressing His-tagged full-length *IFT70* and its various deletion mutants (IFT70*-His) were mixed, respectively, with cell lysates from cells expressing GST-IFT52 and IFT88-His followed by GST pull-down assay. His and GST antibodies were used for immunoblotting. (B) Expression of C-terminal deletion mutant of *FLS2* in *fls2* cells. *fls2* cells expressing HA-tagged full-length (FL) *FLS2* or its C-terminal deletion mutants (ΔCT) (three strains 11, 109, 191) were analyzed by immunoblotting. *fls2* cells were used as a negative control. (C) The C-terminal region of FLS2 is required for its ciliary transport. Cilia were isolated from *fls2* cells expressing full-length *FLS2* or ΔCT mutant that were treated with or without NaPPi for 10 min followed by immunoblot analysis. (D) The C-terminal region of FLS2 does not affect its kinase activity. FLS2 was immunoprecipitated with anti-HA from cell samples as indicated and subjected to immunoblot analysis and *in vitro* kinase assay. *In vitro* kinase assay was performed as shown in Fig 3C. (E-F) Failed ciliary transport of FLS2 by C-terminal deletion induces CrKinesin13 phosphorylation and impairs ciliary disassembly. Cilia isolated from cell samples as indicated were analyzed by immunoblotting (E). *fls2* cells expressing full-length (FL) *FLS2* or ΔCT mutant were induced for ciliary disassembly by NaPPi treatment followed by ciliary length measurement at the indicated times. *fls2* cells were used as control. Bars indicate SD.

control proper phosphorylation of CrKinesin13 in cilia. This study reveals an active role of IFT in regulating ciliary disassembly (resorption) triggered by internal and external cues.

IFT is well established for its function in ciliary assembly and maintenance [27, 28, 48]. However, whether IFT actively functions in ciliary disassembly is not clear. Inactivation of IFT motor kinesin-2 in temperature sensitive mutants of *Chlamydomonas* induces deciliation as well as ciliary shortening [23, 29, 31]. Interestingly, acute ablation of kinesin-2 in mammalian induces ciliary disassembly mainly via deciliation [30]. It is not clear how inactivation of kinesin-2 triggers distinct modes of ciliary disassembly. Nevertheless, these results suggest that down-regulation of IFT may be one of the mechanisms for ciliary disassembly. However, during ciliary resorption triggered by extracellular stimuli or under physiological conditions in *Chlamydomonas*, increased ciliary transport of IFT proteins was observed [25, 32, 33], demonstrating that up-regulation instead of down-regulation of IFT is related to ciliary disassembly. Previously, it has been shown that CrKinesin13 is transported to cilia via IFT during ciliary disassembly [25]. However, it fails to show whether disruption of this transport would impair ciliary disassembly. Our finding that FLS2 is transported by IFT70 to cilia and interruption of

this transport impairs ciliary disassembly provides direct evidence for an active role of IFT in ciliary disassembly.

Ciliary resorption entails depolymerization of the axonemal microtubules, which are the backbones of cilia. CrKinesin13 is timely transported to cilia to mediate depolymerization of axonemal microtubules upon induction of cilia resorption [13, 25].

Kinesin13 members KIF2A and KIF24 in mammalian cells are also implicated in cilia resorption [20, 24, 26]. However, they were shown to be localized in the basal body region, raising the question of how they mediate ciliary disassembly that occurs at the ciliary tip [49]. One may speculate that mammalian kinesin13s may also be transported to cilia during ciliary resorption.

In *Chlamydomonas*, kinesin13 in the cilium becomes partially phosphorylated during later stages of ciliary disassembly [13]. As phosphorylation of CrKinesin13 down-regulates its activity [50], it is proposed that later onset of CrKinesin13 phosphorylation is to ensure constant rate of disassembly because of the polarized nature of the cilium (see discussion in [13]). The regulation of CrKinesin13 phosphorylation is not clear. We have shown here that FLS2, a CDK-like kinase, is timely transported to cilia in a similar manner to CrKinesin13 upon induction of ciliary disassembly. Loss of FLS2 or abrogation of its ciliary transport induces early onset of CrKinesin13 phosphorylation in cilia, suggesting that FLS2 functions in suppressing CrKinesin13 phosphorylation. The later onset of CrKinesin13 phosphorylation in wild type cells can be explained by gradual dephosphorylation and inactivation of FLS2 during ciliary disassembly.

Previously, we have shown that FLS1, another CDK-like kinase, is involved in ciliary disassembly [13]. However, these two kinases exhibit distinct modes of regulation during ciliary disassembly. FLS1 is present both in the cell body and cilia. The cell body form of FLS1 is phosphorylated and activated upon induction of ciliary disassembly to regulate CALK phosphorylation while the cilia form is constitutively phosphorylated. In contrast, FLS2 is a cell body protein. It is gradually inactivated and transported to cilia during cilia resorption. Loss of either FLS1 or FLS2 misregulates CrKinesin13 phosphorylation, suggesting that FLS1 and FLS2 act non-redundantly but collaboratively suppress CrKinesin13 phosphorylation. As FLS1 and FLS2 do not affect each other, they are not in a linear signaling cascade to regulate kinesin13 phosphorylation. The direct targets of FLS1 and FLS2 remain to be identified.

Our finding that FLS2 is involved in ciliary disassembly and cell cycle progression may have implications for functions of human CDKLs in brain development and etiology of related diseases. Patients with mutations in *CDKL2*, *CDKL3* or *CDKL5* exhibit symptoms in intellectual and developmental disabilities [51, 52]. How can defects in ciliary disassembly cause neuronal developmental disorders? Ciliary disassembly is linked with G1-S transition [17, 18]. Mutated centrosomal-P4.1-associated protein (CPAP) or disruption of WDR62-CEP170-KIF2A pathway causes long cilia, retarded ciliary disassembly, and delayed cell cycle re-entry, leading to premature differentiation of neural progenitors [20, 21]. Similarly, Tctex-1 also functions in ciliary disassembly and fate determination of neural progenitors [18]. Thus, it is likely that mutations in CDKLs result in defects in ciliary disassembly and cell cycle progression in the brain, leading to mal-differentiation of neural cells.

## Materials and methods

### Strains and culture conditions

*Chlamydomonas reinhardtii* strains *21gr* (mt+; CC-1690) and *6145c* (mt−; CC-1691), available from the *Chlamydomonas* Genetic Center (University of Minnesota). *fls1* and *fls2* mutants (both in *21gr* background) were generated in previous and current studies [13], respectively.

Cells were cultured in liquid R medium followed by growth in liquid M medium for 2–3 days at 23˚C with aeration under a light/dark (14/10 h) cycle, as reported previously [53]. To induce gamete differentiation, cells were incubated in nitrogen free M medium for 20 hrs under continuous light. Equal number of gametes of opposite mating types were mixed to allow zygote formation.

### Cell cycle analysis

Cells were synchronized by growth in M medium in light/dark (14/10 h) cycle with aeration of 5% $CO2$ in a Light Incubator (Percival AL-36, USA). Cell density was maintained between $10^5$ cells/ml and $10^6$ cells/ml by dilution into fresh M media at the beginning of each light phase [54]. Cells at the indicated times were fixed with 1% glutaraldehyde followed by scoring cell division microscopically.

### Cilia isolation, fractionation and ciliary disassembly

Cilia were isolated after deflagellation by pH shock followed by sucrose gradient purification and fractionated into membrane matrix and axonemal fractions by using 1% NP40 [55]. To induce ciliary disassembly *in vitro*, cell cultures were treated with 20 mM of sodium pyrophosphate (final concentration) for 10 min or indicated time [23, 34]. To examine ciliary disassembly during zygotic development, cells of opposite mating types were induced to undergo gametogenesis in M medium without nitrogen for 20 hrs under continuous light, respectively, followed by mixing to allow mating, zygote formation and development [32]. For zygote formation of cells in *fls2* background, *fls2* (mt+) was crossed with *6145C* (mt-) to generate an mt-*fls2* progeny. Cells were fixed with 1% glutaraldehyde at the indicated times followed by bright field imaging via an inverted Zeiss microscope (Axio Observer Z1, Zeiss) equipped with a charge-coupled device (CCD) camera using a 40 x objective and processed in ImageJ (NIH) and GraphPad Prism (GraphPad, USA). Ciliary length was measured from at least 50 cells. The results are represented as mean ± s.d. For statistical analysis, unpaired two-tailed *t* test was used.

### Insertional mutagenesis and transgenic strain generation

*fls2* was generated by transformation of *21gr* with a 2.1 kb DNA fragment containing the paromomycin resistance gene *AphVIII* [56]. The disrupted gene was identified by cloning the flanking genomic sequences using RESDA PCR followed by DNA sequencing [56]. To make a construct for expressing *FLS2-HA* in *fls*2 mutant, a full length genomic clone of *FLS2* with endogenous promoter was obtained by PCR from a bacterial artificial chromosome (BAC) containing the *FLS2* gene (BAC 34I11, Clemson University Genomics Institute). A 3xHA tag sequence followed by a Rubisco terminator was cloned from plasmid pKL-3xHA (kindly provided by Karl F. Lechtreck, University of Georgia). The resulting construct was cloned into a modified vector pHyg3 that harbors a hygromycin B resistance gene [57]. *FLS2* deletion and *K33R* mutants were constructed based on the wild type *FLS2* gene construct. The final constructs were linearized with XbaI and transformed into the *fls2* mutant using electroporation.

## RT-PCR

Total RNA was isolated (HiPure Plant RNA MIni Kit, Magen) and reverse transcribed (The PrimeScript® RT reagent Kit, Takara). Two pairs of primers (5′-GGCAAGAATCATCACGACCAG-3′ and 5′-GTATGCCATGAGGTCGTCCAC-3′) for *FLS2*, and (5′-ATGTGCTGTCCG

TGGCTTTC-3′ and 5′-GCCCACCAGGTTGTTCTTCA-3′) for *CBLP*, were used for PCR. The amplified fragments were subjected to 1.5% agarose gel electrophoresis.

## Gene silencing by RNAi

Knockdown of *FLS1* expression in *fls2* mutant was performed by using artificial microRNA (amiRNA) essentially as previously described [58]. The targeted DNA sequence was selected following the instruction in http://wmd3.weigelworld.org. Synthesized oligonucleotides for amiRNA after annealing were cloned into the p3int-RNAi vector at the SpeI/NheI restriction sites [59]. The final construct was linearized with ScaI and transformed by electroporation into *Chlamydomonas*. The targeted sequence: TGCAGGAGCTTAATGTCGCGC. The oligonucleotides: FLS1-F1, CTAGTGCGCGACATTAAGCTTTTGCATCTCGCTGATCGGCACCAT GGGGGTGGTGGTGATCAGCGCTATGCAGGAGCTTAATGTCGCGCG; FLS1-R1, CTAG CGCGCGACATTAAGCTCCTGCATAGCGCTGATCACCACCACCCCCATGGTGCCGA TCAGCGAGATGCAAAAGCTTAATGTCGCGCA.

## Primary antibodies

The following antibodies were used for immunoblotting (IB) or immunofluorescence (IF). Rat monoclonal anti-HA (1:50 for IF and 1:1000 for IB; Roche); mouse monoclonal anti-$\alpha$-tubulin (1:200 for IF and 1:2500 or 1:10,000 for IB; Sigma-Aldrich); rabbit polyclonal anti-CALK (1:10000, IB) (pan 2004); rabbit polyclonal anti-CrKinesin13 (1:3000, IB) [25]; rabbit polyclonal anti-IFT46 (1:2000, IB) [60]; rabbit polyclonal anti-IFT121 (1:2000, IB) [53]; mouse anti-FMG1 (1:5,000, IB) [61]; mouse polyclonal anti-FLS1 (1:1500, IB) [13] and anti-Thiophosphate (1:5000, IB, Abcam). We attempted to make anti-FLS2 antibody in rabbits and mouse by using 641–841 aa and 756–1106 aa as antigens, respectively, but none were successful.

## Immunoblotting and immunofluorescence

IB and IF experiments were performed as described previously [55]. The secondary antibodies used for IF are the following: Goat anti-Rat IgG Alexa Fluor 488, Goat anti-Mouse IgG Alexa Fluor 594. Cells were analyzed under a Zeiss LSM780 META Observer Z1 Confocal Laser Microscope and the images were acquired and processed by ZEN 2012 Light Edition (Zeiss). The images were processed in Adobe Photoshop and assembled in Adobe Illustrator (Adobe).

## Phosphatase treatment and Phos-tag SDS-PAGE

Cell samples ($5 \times 10^6$ cells) were lysed in 40 µl Buffer (50 mM Tris pH 7.5, 10 mM MgCl$_2$) containing protease inhibitor cocktail and 25 µg/ml ALLN. For phosphatase treatment, the final 50 µl reaction buffer contained 38 µl of cell lysate, 2 µl lambda protein phosphatase (800 U) and buffer components as instructed (Sigma) and the reaction was terminated after 30 min at 30 $^\circ$C. To visualize protein phosphorylation by gel mobility shift, proteins were separated in phos-tag gel system [38]. A 6% SDS-PAGE with 20 mM Phos-tag acrylamide (Wako) was used. After electrophoresis, the divalent cations were removed from the Phos-tag gels by incubating them with transfer buffer containing 2 mM EDTA for 20 min before membrane transferring.

## Immunoprecipitation (IP) and *in vitro* protein kinase assay

$1 \times 10^9$ cells were lysed in IP buffer (20 mM Hepes, pH 7.2, 5 mM MgCl$_2$, 1 mM DTT, 1 mM EDTA, 150 mM NaCl, EDTA-free protease inhibitor cocktail, 25 µg/ml ALLN). For IP with isolated cilia, 1 mg cilia were lysed in IP buffer supplemented with 0.6 M KCl and 0.05 M

$CaCl_2$. The lysates were incubated with 30 μl pre-washed rat anti-HA Affinity Matrix (Roche) for 3 h at 4°C, followed by washing three times with IP buffer containing 0.1% NP40 and 0.1% Triton X-100 and centrifugation. For *in vitro* protein kinase assay, it was performed with ATPγS as a phosphodonor and anti-thiophosphate ester antibody to detect substrate phosphorylation [62]. The immunoprecipitates of FLS2-HA were incubated at room temperature for 30 min in 30 μl reaction buffer (10 mM HEPES pH 7.2, 150 mM NaCl, 10 mM $MgCl_2$, 5 mM DTT, 2 μg MBP and 1 mM ATPγS (Abcam)), followed by addition of 1.5 μl of 50 mM p-nitrobenzylmesylate (Abcam) for 2 h. Protein phosphorylation was detected by immunoblotting with anti-thiophosphate ester antibody.

### Yeast-based two-hybrid analysis

cDNAs of *FLS2* and IFT genes cloned by PCR from a *Chlamydomonas* cDNA library were cloned into yeast expression vectors pGBKT7 and pGADT7, respectively. The resulting constructs were co-transformed into AH109 yeast cells with different combinations. The transformants were grown at 30 °C for 2–3 days on selection medium SD lacking leucine, tryptophan, histidine, and adenine (SD, -Leu, -Trp, -His, -Ade) or lacking leucine and tryptophan (SD, -Leu, -Trp).

### Pull-down assays

$GST$-$FLS2$-$CT_{290-1106}$ and $GST$-$IFT52$ were cloned into bacterial expression vector pGEX-6P-1, respectively. IFT88-His, IFT70-His and various His-tagged IFT70 deletion mutants were cloned into bacterial expression vector pET28a, respectively. The proteins were expressed in BL21 cells. For pull-down assay of GST-FLS2-CT and IFT70-His, these two proteins were co-expressed. For pull-down assays of IFT70 or its variants with IFT88-His and GST-IFT52, IFT70 and its variants were separately expressed while IFT88-His and GST-IFT52 were co-expressed. The cell lysates were mixed as indicated followed by pull-down and immunoblotting.

## Supporting information

**S1 Table. Numerical data for graphs.**
(XLSX)

## Acknowledgments

We are grateful to Drs. Robert Bloodgood, Kaiyao Huang and Karl Lechtreck for providing antibodies and/or plasmids. We thank previous laboratory members for preliminary data. We also thank the Core Facility of the Center for Biomedical Analysis (Tsinghua University) for assistance on cell imaging analysis.

## Author Contributions

**Conceptualization:** Junmin Pan.

**Data curation:** Qin Zhao, Junmin Pan.

**Formal analysis:** Qin Zhao, Shufen Li, Shangjin Shao, Zhengmao Wang, Junmin Pan.

**Funding acquisition:** Junmin Pan.

**Investigation:** Qin Zhao, Shufen Li, Shangjin Shao, Zhengmao Wang.

**Methodology:** Qin Zhao, Shufen Li, Shangjin Shao, Zhengmao Wang.

**Project administration:** Junmin Pan.

**Resources:** Junmin Pan.

**Software:** Junmin Pan.

**Supervision:** Junmin Pan.

**Validation:** Qin Zhao.

**Visualization:** Qin Zhao.

**Writing – original draft:** Qin Zhao, Junmin Pan.

**Writing – review & editing:** Qin Zhao, Junmin Pan.

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
