## [Decision Letter · Decision Letter 0]

22 Jan 2020

Dear Junmin and colleagues

Thank you very much for submitting your Research Article entitled 'FLS2 is a CDK-like kinase that directly binds IFT70 and is required for proper ciliary disassembly in Chlamydomonas' to PLOS Genetics. Your manuscript was fully evaluated at the editorial level and by independent peer reviewers. The reviewers appreciated the attention to an important topic but identified some aspects of the manuscript that should be improved.

We therefore ask you to modify the manuscript according to the review recommendations before we can consider your manuscript for acceptance. Your revisions should address the specific points made by each reviewer.  In particular, it is important that you address the comment from reviewer 2 about the discrepancy between the reported effects of fls1 and fls2 mutations in this and a previous publication from this lab. Unlike in the previous publication, fls1 shows a constant slow rate of disassembly.  At the very least this should be acknowledged and discussed.

[LINK]

Yours sincerely,

Susan K. Dutcher

Associate Editor

PLOS Genetics

Gregory P. Copenhaver

Editor-in-Chief

PLOS Genetics

Reviewer's Responses to Questions

**Comments to the Authors:**

Reviewer #1: Review uploaded as attachment

Reviewer #2: Zhao et al. describe a new Chlamydomonas mutant, fls2, that alters ciliary disassembly rates. The disrupted gene encodes a CDK-like protein kinase, and the authors explore several potential mechanisms for the involvement of this kinase in regulation of ciliary disassembly during physiologically controlled disassembly events, including zygote formation and the cell cycle. This topic has been difficult to approach and the use of genetics here appears to provide a useful way forward on an interesting topic.

Extensive data are present that clearly show FLS2 is present in the cytoplasm and at very low levels in cilia, that ciliary abundance increases during disassembly, and that FLS2 is a phosphoprotein that gradually becomes dephosphorylated during disassembly. Unlike some previously analyzed kinases that control ciliary length, during disassembly FLS2 appears to be distributed evenly along the length of the cilia, and is associated with axonemal structures following detergent treatment.

Based on previous work from this and other labs, the effects of this mutation on other proteins implicated in disassembly, including a microtubule-depolymerizing kinesin13 and an aurora kinase (CALK) were also tested. Unlike a previously analyzed mutation in a closely related CDKL, FLS1, the fls2 mutation did not alter the disassembly-linked pattern of CALK phosphorylation, whereas changes to the pattern of kinesin13 phosphorylation were similar in fls2 and fls1.

Finally, the authors provided strong support for their contention that trafficking of FLS2 into cilia in response to a disassembly signal is IFT-dependent and involves an interaction between the non-kinase portion of FLS2 and several TPR domains in IFT70.

Overall, this is an interesting examination of the role of this protein kinase in the regulation of ciliary disassembly. Although a direct signal cascade could not be uncovered, the authors should be applauded for the care with which they tested multiple possible mechanisms, including the use of kinase-dead and truncated versions, direct tests of kinase activity, interaction studies using both in vitro and yeast 2-hybrid approaches, and localization studies over extended time frames.

A few aspects of this manuscript should be clarified prior to publication.

1. Perhaps most important among this is a discrepancy between the reported effects of fls1 and fls2 mutations in this and a previous publication from this lab. In their prior paper on fls1, much was made of the kinetics of disassembly, and specifically fls1 mutant cells showed a biphasic disassembly rate, with the initial rate being much slower than wild type and at later times there was a shift to an identical rate to wild type. Here, fls2 displays a constant rate of disassembly that is slower than wild type, suggesting that its effects can be distinguished from those of fls1. However, in Fig. 4I a direct comparison shows no difference between the disassembly kinetics of fls1 and fls2. Unlike in the previous publication, fls1 shows a constant slow rate of disassembly. At the very least this should be acknowledged and discussed.

2. In lines 234-236, the authors state that “The increased amounts of FLS2-HA in cilia between 10 and 120 min after induction of ciliary disassembly was similar, which is indicative of increased ciliary trafficking of FLS2 rather than ciliary accumulation of FLS2 (Fig. 5E).” I agree that the levels are similar, but do not understand why this is indicative of increased trafficking. Since the ciliary FLS2 does not wash out with detergent (Fig. 5C), it is associated with axonemes and not just with the increased amounts of trafficking IFT complexes. Therefore this seems to be a very rapid assembly process, targeting FLS2 to sites distributed along the length of the axoneme. FLS2 does not accumulate when trafficking was blocked by disruption of a temperature-sensitive kinesin (Fig. 6A), but no data is presented to show that this increase in FLS2 could not have occurred without the increase in IFT trafficking.

3. The data presented do overall support a direct interaction of FLS2 and IFT70, however, co-immunoprecipitation of IFT70 with FLS2 from cell or ciliary extracts (Fig. 6E, described in lines 271-272 and in the figure legend) only indicates that these two proteins interact within a complex, not that they interact directly with each other in vivo.

**Have all data underlying the figures and results presented in the manuscript been provided?**

Reviewer #1: Yes

Reviewer #2: Yes

PLOS authors have the option to publish the peer review history of their article (what does this mean?). If published, this will include your full peer review and any attached files.

Reviewer #1: No

Reviewer #2: No

---

## [Editor Report · Decision Letter 1]

1 Feb 2020

Dear Dr Pan,

We are pleased to inform you that your manuscript entitled "FLS2 is a CDK-like kinase that directly binds IFT70 and is required for proper ciliary disassembly in Chlamydomonas" has been editorially accepted for publication in PLOS Genetics. Congratulations!

Yours sincerely,

Susan K. Dutcher

Associate Editor

PLOS Genetics

Gregory P. Copenhaver

Editor-in-Chief

PLOS Genetics

Comments from the reviewers (if applicable):

**Data Deposition**

http://datadryad.org/submit?journalID=pgenetics&manu=PGENETICS-D-19-02009R1

**Press Queries**

---

## [Editor Report · Acceptance letter]

26 Feb 2020

PGENETICS-D-19-02009R1 

FLS2 is a CDK-like kinase that directly binds IFT70 and is required for proper ciliary disassembly in Chlamydomonas 

Dear Dr Pan, 

We are pleased to inform you that your manuscript entitled "FLS2 is a CDK-like kinase that directly binds IFT70 and is required for proper ciliary disassembly in Chlamydomonas" has been formally accepted for publication in PLOS Genetics! Your manuscript is now with our production department and you will be notified of the publication date in due course.

With kind regards,

Kaitlin Butler

PLOS Genetics

On behalf of:
